# Forced Circulation of Nitrogen Gas for Accelerated and Eco-Friendly Cooling of Metallic Parts

**Zu Seong Park [1], Jeong Kim [2], Young Yun Woo [3], Habeom Lee [3], Ji Hoon Kim [3] and Young Hoon Moon [3,*]**

[1]  R&D Center, Poongsan, 2606-10 Hokook-ro, Angang, Gyeongju, Gyeongsangbuk-do 38026, Korea
[2]  Department of Aerospace Engineering, Pusan National University, 30 Jangjeon-dong, Geumjeong-gu, Busan 46241, Korea
[3]  School of Mechanical Engineering, Pusan National University, 30 Jangjeon-dong, Geumjeong-gu, Busan 46241, Korea
*  Correspondence: yhmoon@pusan.ac.kr; Tel.: +82-51-510-2472

**Abstract:** As nitrogen is nonreactive and non-flammable, it can provide a quick and simple medium of cooling and environment protection. One disadvantage of nitrogen cooling is its lower heat transfer coefficient than water. Despite its lower cooling capacity, nitrogen cooling can produce cleaner products, thereby eliminating the need to wash the parts and dispose of the contaminated water. In this study, an innovative nitrogen cooling system was developed for an accelerated and eco-friendly cooling of metallic parts. The dry nitrogen gas, transported via the nozzle of a cryogenic liquid nitrogen tank, is passed into the cooling chamber and exchanges heat with the workpiece. The heated nitrogen gas is forcibly transported to the chiller, where the heat is reduced, and the cooled gas is circulated again. The performance of this nitrogen cooling system has been evaluated with cooling experiments of sintered tungsten heavy alloys. The nitrogen-cooled product shows a clean surface with improved mechanical properties. Furthermore, nitrogen induces less distortion compared to water cooling, thus reducing the post-machining costs.

**Keywords:** nitrogen cooling; forced cooling; distortion; chiller; cooling rate; tungsten heavy alloy

## 1. Introduction

An innovative nitrogen cooling system has been developed for accelerated and eco-friendly cooling of metallic part. In this cooling process, the dry gaseous nitrogen is rapidly circulated and exchanges heat with the workpiece that it contacts, resulting in an accelerated cool-down. The heated nitrogen gas is forcibly circulated and transported to the chiller in a closed loop, where the heat was removed, and the cooled gas was circulated again.

Selection of a cooling medium is mainly based on chemical composition, dimension, and desired surface quality of the parts [1–3]. The function of the cooling medium is to control the rate of heat transfer from the surface of the quenched parts, and the most commonly used cooling media is water. However, the surface oxidation, environmental contamination, and thermal distortion during water cooling have been problematic in industrial applications [1,2].

Argon and nitrogen are the gases that are most utilized for eco-friendly cooling during thermal processing in vacuum furnaces. Gases with smaller and lighter molecules exhibit a greater thermal conductivity, because they can move faster and farther without a collision [1,4,5]. As nitrogen of atomic weight 14.0067 amu is approximately 2.9 times lighter than argon of atomic weight 39.948 amu, it is approximately four times faster than argon [6,7]. Consequently, nitrogen exhibits a greater heat transfer coefficient than argon. Furthermore, the heat transfer coefficient of nitrogen can be significantly

increased by controlling the temperature and pressure [7–10]. As nitrogen is nonreactive, the inert environment created by nitrogen prevents the occurrence of combustible reactions. As nitrogen leave behind no residues on workpieces or in a chamber space, these properties make it unnecessary to invest in washing facilities and fire monitors and lower the operating cost, since maintenance procedures and disposal of the quenching media are eliminated.

Several studies have been performed on the nitrogen cooling process [11–20]. To enhance the cooling rate under operating conditions, liquid nitrogen cooling has been performed. Investigations were carried out in a drilling operation under flood and liquid nitrogen cooling separately. The experimental results indicated a reduction in cutting temperature, increase in thrust force, and surface roughness when liquid nitrogen cooling is applied [11]. The machinability characteristics of turning using dry turning and liquid nitrogen cooling methods has also been investigated. The experimental results indicated improved cutting tool performance under liquid nitrogen cooling by control of wear mechanisms which, in turn, reduced the wear rate [12].

High-pressure gas cooling can be a form of heat treatment process used for hardening of tool and die steels. The use of gas quenching can significantly improve the mechanical and physical properties of a material for obtaining the near-net shape of metal components. The liquid nitrogen die cooling effect has been estimated on the conventional extrusion process [13]. Results showed a significant impact of the design aspects on the thermal efficiency of the cooling and an important heat removal when the liquid nitrogen cooling is used. Wang et al. [14] simulated the high-pressure gas quenching effect on the cooling of a large H13 die by using computational fluid dynamics.

Laser surface melting is a high-energy surface treatment that allows modification of the microstructure and surface properties. An attempt of laser surface melting with liquid nitrogen-assisted cooling was carried out to obtain a higher cooling rate and improve the surface properties. The experimental results showed a thinner melted layer, a highly homogeneous, refined melted microstructure [15]. Manikandan et al. [16] reduced microsegregation by enhancing the weld cooling rate using liquid nitrogen cooling during the gas tungsten arc welding process. Additionally, as the liquid nitrogen cooling increased the fracture degree inside the solids, liquid nitrogen has also been used to bring about thermal damage cooling as a fracturing fluid [17,18].

As described above, the primary efforts have been mainly focused on discovering appropriate approaches to generate sufficiently high cooling efficiency through the direct contact of cryogenic liquid nitrogen [11–18] and produce cooling that results in uniform mechanical properties of the products [19,20]. However, practical applications of dry nitrogen gas in the cooling process of metallic parts have not been sufficiently reported [1–5,7–10]. Gases have a fundamental disadvantage in comparison to fluids, which is that they have poor heat transfer characteristics under normal condition. For their use as cooling media in cooling system, they have to be optimized by proper adjustment of the gas pressure and flow speed.

In this study, the nitrogen gas cooling system has been developed to exchange heat with the workpieces, resulting in an accelerated and eco-friendly cool-down. The heated nitrogen gas is forcibly circulated and transported to the chiller in a closed loop, where the heat is removed, and the nitrogen gas is cooled. This cooled nitrogen gas is rapidly circulated again using a circulation pump rotating at a high speed.

To estimate the performance of the proposed nitrogen cooling system, sintered tungsten heavy alloy (THA) was cooled using nitrogen gas. It has been observed that the mechanical properties of the sintered THA parts have significantly improved after the heat treatment. Especially, accelerated cooling after heat treatment was advantageous for the improved quality of THA parts [21–23]. THAs are composites consisting of nearly spherical tungsten particles embedded in a ductile matrix phase with lower melting point elements, such as nickel, iron, copper, and cobalt [24,25]. As tungsten melts at excessively high temperatures, a mixture of tungsten powders with a minimal amount of a lower-melting-point metal was used to form a liquid phase when heated to a moderate temperature. The lower-temperature melting elements were considered so that the liquid phase could dissolve

some tungsten in the solution, thereby inducing wetting of the tungsten grains. The THA is usually processed using powder technology by employing the liquid phase sintering (LPS) process [26,27]. This process is suitable for alloys that melt over a range of temperatures. It can be conducted under conditions, in which solid grains coexist with a wetting liquid. As the final product is a multi-phase with customized properties, LPS is considered one of the most dominant commercial sintering processes. THAs are widely used in manufacturing radiation shielding and some parts for the military defense, such as missile weapons and armor piercing ammunition, in tubular or hollow shapes. In general, tubular parts are fabricated by conventional metal forming processes. Rotary swaging [28,29] and roll forming [30] methods are usually applied to manufacture tubular parts. For more complicated tubular products, hydroforming [31–33] and electromagnetic forming [34] are used. In general, the excessively high consolidation temperature of tungsten does not allow the application of solid-state metal forming method to fabricate tubular parts. Thus, one possible alternative to handle this situation is to combine the segmented THA pieces by solid-state diffusion bonding to build up a tubular component. As the sintering process is required for fabricating the THA tubular part, controlling the sintering temperature is very important. Accurate measurement and control of temperature during high-temperature processes are important because it affects the product quality and operational performance [35–42].

An extremely dynamic and rapid cooling process does not permit a stable and consistent measurement of cooling rates. Therefore, finite element analysis was conducted to estimate the overall heat transfer coefficient of the developed nitrogen cooling system. Furthermore, the economical comparison of the nitrogen cooling system has been performed. To estimate the performance of the proposed nitrogen cooling system, the mechanical properties, distortion, and surface cleanliness of the THA parts have also been compared with that of water cooling.

## 2. Materials and Methods

### 2.1. Materials

The tubular workpiece used in this study was made of THA having 5 wt%Ni–2 wt%Fe binder metals. The as-received powders were de-agglomerated using a vibrating 200 mesh screen. The elemental powders of W, Ni, and Fe were weighed according to the target compositions and blended using a 20-rpm blender for 2.5 h to achieve a homogeneous distribution. The blended tungsten alloy powders were ball-milled for 5 h in a jar filled with 1.0–1.5 inch balls under an argon environment to break up the agglomerates, provide rigorous mixing, and uniformly disperse the elemental powders [43,44]. After the ball milling, the blended powders are crushed again using a Fitz mill, which comminutes the powders with blades rotating at 3450 rpm. After the Fitz milling process, the powders were screened using a vibrating 150 mesh screen and blended again by a 20-rpm blender for 1.5 h to obtain a homogeneous powder. The THA powders are difficult to compact due to their high hardness and tendency of minimal deformation under pressure [21]. In this study, milled powders were placed in a flexible rubber mold and subjected to hydrostatic pressure for achieving a more uniform compaction and therefore density. The assembly was then pressurized hydrostatically in a chamber under 195 MPa hydrostatic pressure for 20 s. A tubular powder compact was prepared using the cold isostatic pressing (CIP) method, and liquid phase sintered. Figure 1 shows the process sequence adopted for fabricating the THA tube through the LPS process. The $D_i$, $D_o$, and $L$ of the tubular workpiece are 0.168 m, 0.204 m, and 0.210 m, respectively.

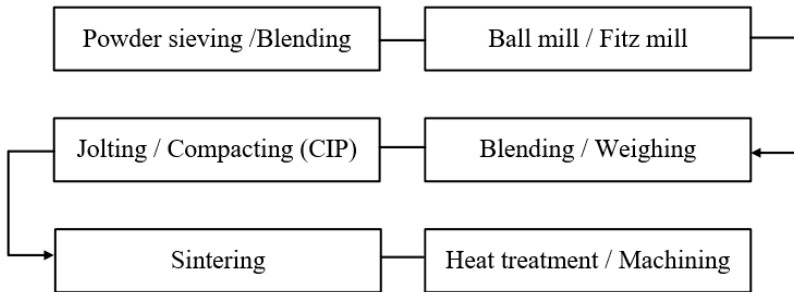

**Figure 1.** Process sequence for the THA tube fabrication.

The microstructural evolution and mechanical properties of sintered THAs can be adjusted using the post-heat treatment. Particularly, the cooling rate of post-heat treatment must be controlled to enhance the mechanical properties of THSs [24,25]. A slow cooling in the furnace to room temperature could result in an enhanced growth of tungsten (W) spheroids, determining the decrease of mechanical properties of THAs [21–23]. Therefore, faster cooling is advantageous to enhance mechanical properties of sintered THAs. In this study, sintered THA has been heat treated and cooled by nitrogen gas to estimate the performance of the proposed nitrogen cooling system.

*2.2. Methods*

2.2.1. Design of the Dry Nitrogen Cooling System

Figure 2 shows the nitrogen cooling system developed in this study. The gaseous nitrogen, which was transported via a specially-designed nozzle of the cryogenic liquid nitrogen tank, is passed into the cooling chamber. As the surge tank with a volume of 15,000 L was filled with nitrogen under a pressure of 8.0 bar, nitrogen could be rapidly filled in the cooling zone. In this study, the cooling chamber was controlled to be filled with nitrogen at a pressure of 5 bar. This nitrogen gas exchanged heat with the workpieces it contacted, resulting in an accelerated cool-down. Cooling of the workpiece resulted from the forced convective heat transfer between the workpiece surface and nitrogen.

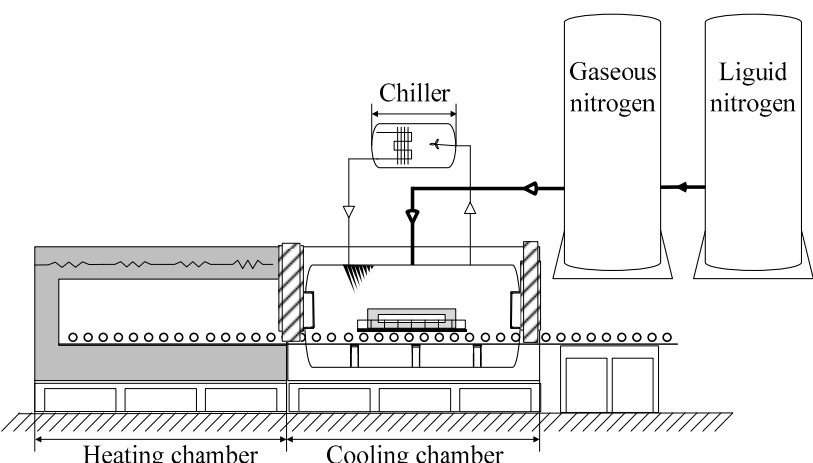

**Figure 2.** Schematic drawing of the proposed nitrogen cooling system.

As shown in Figure 3, the heated nitrogen gas is forcibly circulated and transported to the chiller in a closed loop, where the heat is removed, and the nitrogen gas is cooled. This cooled nitrogen gas is rapidly circulated again using a circulation pump rotating at a speed of 1000 rpm.

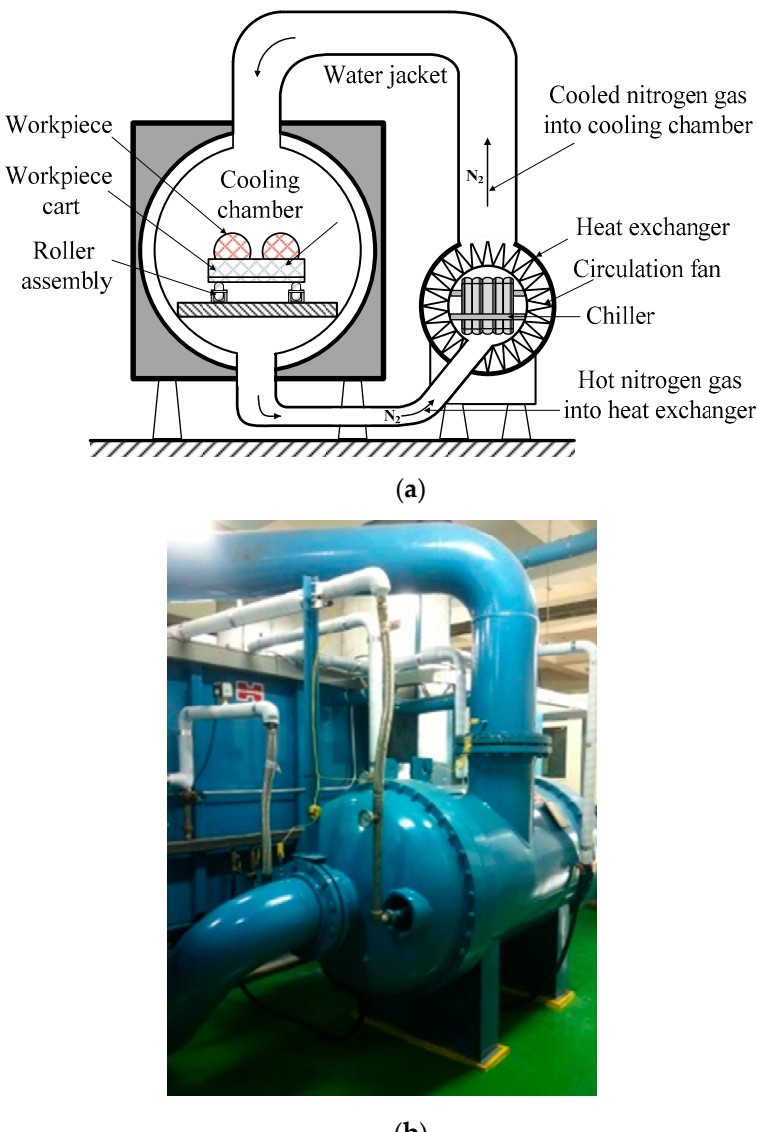

(a)

(b)

**Figure 3.** (**a**) Schematic drawing of closed loop nitrogen circulation; (**b**) photograph of the developed nitrogen cooling system.

Figure 4 presents the operating sequence of furnace with the nitrogen cooling system. Figure 4a shows the workpiece before the heat treatment. For heat treatment, workpieces are loaded in heating zone and the loading gate is closed. The workpiece positioned at the heating zone is heated under a nitrogen atmosphere, as shown in Figure 4b. Then, the heat-treated workpieces are transported to the nitrogen cooling zone and cooled by nitrogen, as shown in Figure 4c.

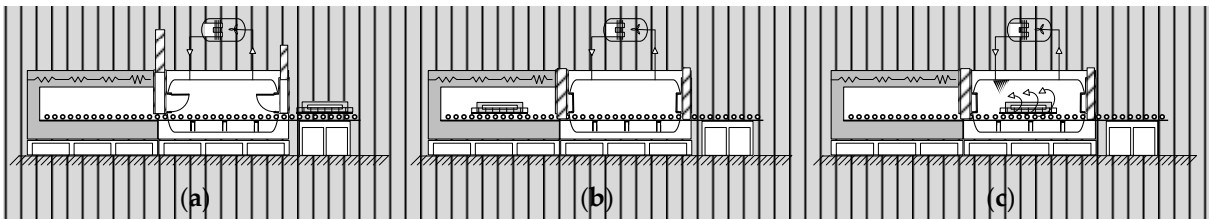

(a)　　　　　　　　　　　(b)　　　　　　　　　　　(c)

**Figure 4.** Operating sequence of the furnace with the nitrogen cooling system: (**a**) Before workpiece loading; (**b**) workpiece loading and heating; (**c**) nitrogen cooling.

As the environmental conditions of the heating and cooling zones must be precisely controlled, the door between the heating and cooling zones as well as the external door is specially designed to prevent leakage [45]. Figure 5 shows the developed door system for the high-quality process cooling. A roller-type moving system is designed to enable faster movement of the workpiece. The time to move the workpiece from the heating chamber to cooling chamber is less than 15 s.

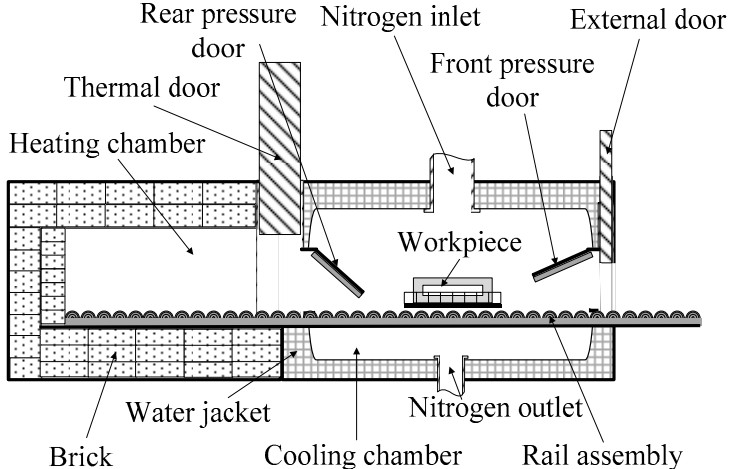

**Figure 5.** Schematic drawing of the door system designed for the cooling process.

2.2.2. Application of the Nitrogen Cooling System to Cool the Sintered THA

The characteristics of nitrogen cooling were compared with those of water cooling. The sintered THA was heat-treated at 1150 °C for 8 h and then cooled using water as well as nitrogen cooling systems. Figure 6 shows the heat treatment cycle performed in this study.

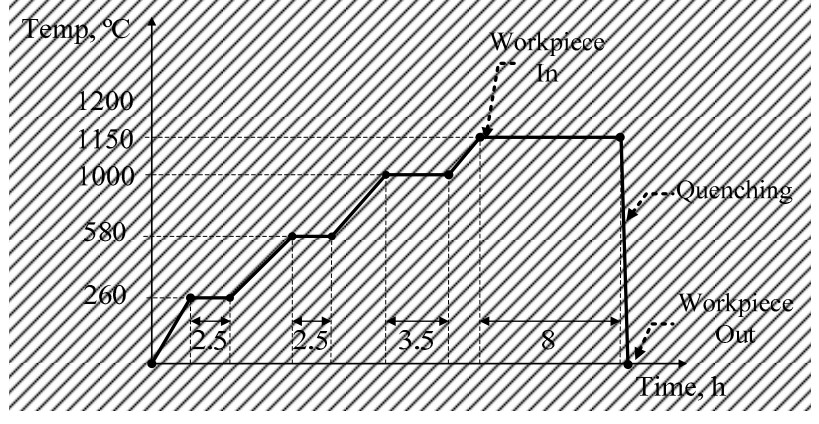

**Figure 6.** Heat treatment cycle.

Conventional water cooling system mainly includes a heating zone and a water-quenching zone. In the heating zone, the workpieces are heated under controlled atmosphere. In the water-quenching zone, the heat-treated products are rapidly cooled in the water pool.

Figure 7 shows the furnace with the water cooling system used in this study. Figure 7a shows the workpiece before heat treatment. For performing the heat treatment, the workpieces are loaded in the heating zone, and the loading gate is closed. The workpiece positioned in the heating zone is heated to 1150 °C under controlled atmosphere, as shown in Figure 7b. Then, the heat-treated workpieces are moved to the water-quenching zone and cooled by water, as shown in Figure 7c. Cooling in the water pool is continued, until the temperature of the product attains room temperature. Then the product is extracted from the water pool and air-dried.

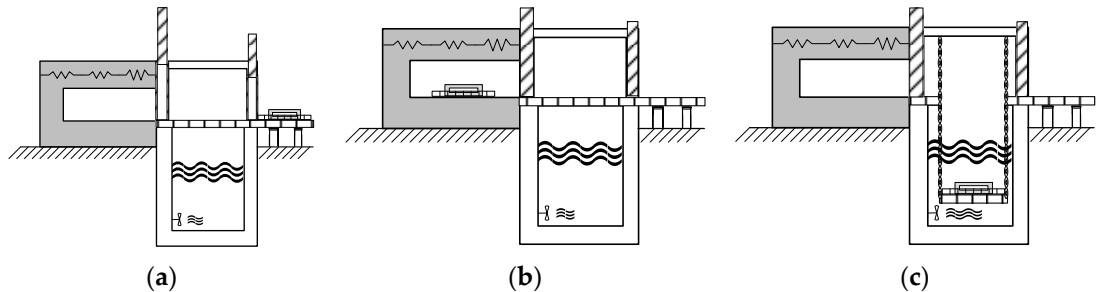

**Figure 7.** Schematic drawing of the furnace with the water cooling system: (**a**); Before workpiece loading (**b**) workpiece loading and heating; (**c**) water cooling.

An extremely dynamic and rapid cooling process does not permit a stable and consistent measurement of cooling rates. Furthermore, as each configuration suffers from non-uniform cooling rates in the longitudinal and circumferential directions, and an experimental measurements of reliable cooling rate is difficult to accomplish. In the case of nitrogen cooling, the cooling of the workpiece progressed in the cooling chamber. Therefore, the cooling of the workpiece was monitored by the temperature of nitrogen in the cooling system. Figure 8 shows the on-line measured temperature profiles of nitrogen gas before and after the heat exchange in the chiller. As shown in this figure, the hot nitrogen gas was rapidly cooled by the heat exchanger and, eventually, the temperatures of the nitrogen gas before and after the heat exchange turned similar after about 2000 s. This similarity in temperature before and after the heat exchange implies that the actual cooling of the workpiece was completed. Therefore, the nitrogen cooling of the workpiece (until it attained the room temperature) was completed after 2000 s. To estimate the overall heat transfer coefficient of the nitrogen cooling system, the transient heat analysis was conducted using the commercial finite element analysis code ABAQUS$^{TM}$ (Abaqus 6.13, Dassault Systems, Providence, RI, USA) [46].

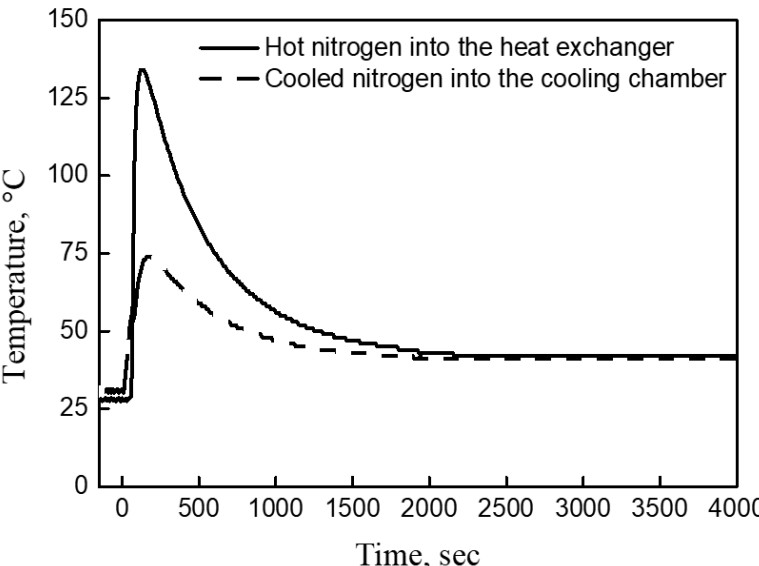

**Figure 8.** Measured nitrogen temperatures before and after passing the heat exchanger.

To analyze the cooling process that employed water and nitrogen gas, the tubular workpiece was modeled with an element dimension of 0.02 mm × 0.02 mm × 0.02 mm, as shown in Figure 9. The $D_i$, $D_o$, and $L$ of the tubular model are 0.4 m, 0.6 m, and 0.5 m, respectively.

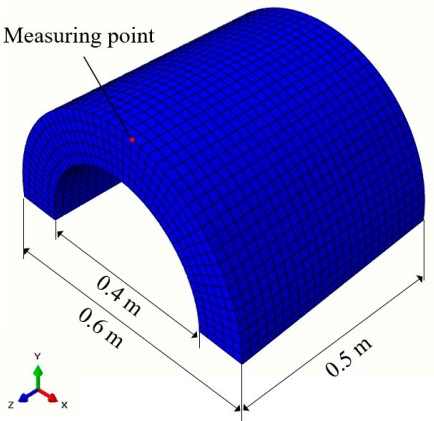

**Figure 9.** Finite element method modeling of the workpiece.

The element type was an eight-node heat transfer brick element (DC3D8). As thermal properties of THA, $\rho$ = 18,500 kg m$^{-3}$, $k$ = 125 W m$^{-1}$ °C$^{-1}$, and $C_p$ = 133 J kg$^{-1}$ °C$^{-1}$ were used in this study. During the cooling process, the heat was removed using three main heat transfer mechanisms, i.e., convection, conduction, and radiation. Varied film coefficients were applied at the surface of specimen for cooling with water and nitrogen gas. A heat transfer coefficient ($h$) value of 580 W m$^{-2}$ °C$^{-1}$ was used for water [46,47] and $h$ = 30, 70, 110, 190, 270, and 350 W m$^{-2}$ °C$^{-1}$ were used as heat transfer coefficients for the nitrogen gas, because the heat transfer coefficient of nitrogen gas varies according to pressure and temperature [2,48]. The initial temperature of the specimen was set to 1150 °C, and the ambient temperature was set to 27 °C. The temperature at the surface of the specimen was obtained, as shown in Figure 9. Figure 10 shows the simulated cooling curves across different heat transfer coefficients. As expected, the cooling rate of water cooling is observed to be higher than that of nitrogen cooling. It is also observed that the cooling rate increases with increasing heat transfer coefficient for nitrogen gas cooling. At the cooling completion time of 2000 s, the heat transfer coefficient of the developed nitrogen cooling system is observed to be approximately 270 W m$^{-2}$ °C$^{-1}$, because the time to attain the room temperature at this value is similar with that of Figure 8. As the heat transfer coefficient of nitrogen in the atmospheric pressure is in the range of 30–50 W m$^{-2}$ °C$^{-1}$, it can be observed that it is significantly elevated due to the developed cooling system that performs a high-speed circulation under high pressure.

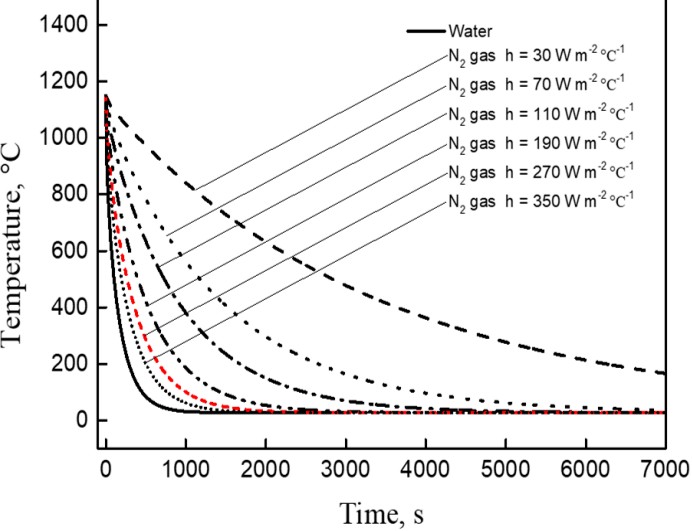

**Figure 10.** Cooling curves with respect to water and nitrogen gas cooling.

## 3. Results and Discussion

### 3.1. Surface Cleanliness

Figure 11 shows the surfaces of the workpieces after the water and nitrogen cooling processes. In the case of water cooling, a thick oxide layer is observed on the surface of the parts, as shown in Figure 11a, while a clean surface is obtained with nitrogen cooling, as shown in Figure 11b. As the inert environment created from nitrogen prevented any oxidation reactions, the surface of the workpiece after cooling was clean without oxidation. This eliminated the need of secondary machining to remove the oxidized surface.

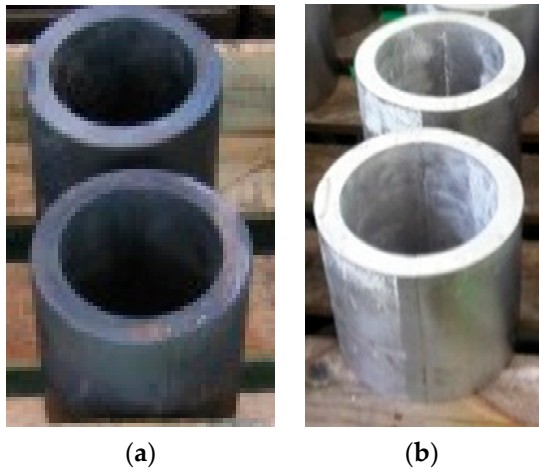

(a)  (b)

**Figure 11.** Surface cleanliness: (**a**) Water cooling; (**b**) nitrogen cooling.

### 3.2. Thermal Distortion

Although cooling at uniform cooling rates in a circumferential direction, longitudinal direction, and inside or outside is ideal, it is practically impossible. Generally, process-induced distortion is generated by non-uniform stresses caused by a non-uniform temperature gradient during the cooling process or non-uniform deformations [49]. In this study, the dimensional accuracies after water and nitrogen cooling were measured. To estimate the degree of distortion, the deviations from the circularity are measured at three different positions. Figure 12 shows the comparison of distortions in tubular parts after water and nitrogen cooling. As shown in this figure, nitrogen gas cooling provides more uniform cooling and less distortion than water cooling, thus reducing post-quenching machining. The standard deviations of distortion for water and nitrogen cooling are 0.343 mm and 0.090 mm, respectively. In the case of water cooling, the differences in cooling rates between the inner and outer tubular surfaces and the locally different immersion times might increase the distortion. In particular, larger distortions are expected in the non-axisymmetric part, cup-shaped part, thin-flat part, and the part with a high length to diameter ratio, where uniform heat transfers cannot occur. However, the distortion in the case of nitrogen cooling is significantly smaller than in the case of water cooling, because nitrogen can cool the parts more uniformly and rapidly due to the forced circulation of the gaseous flow.

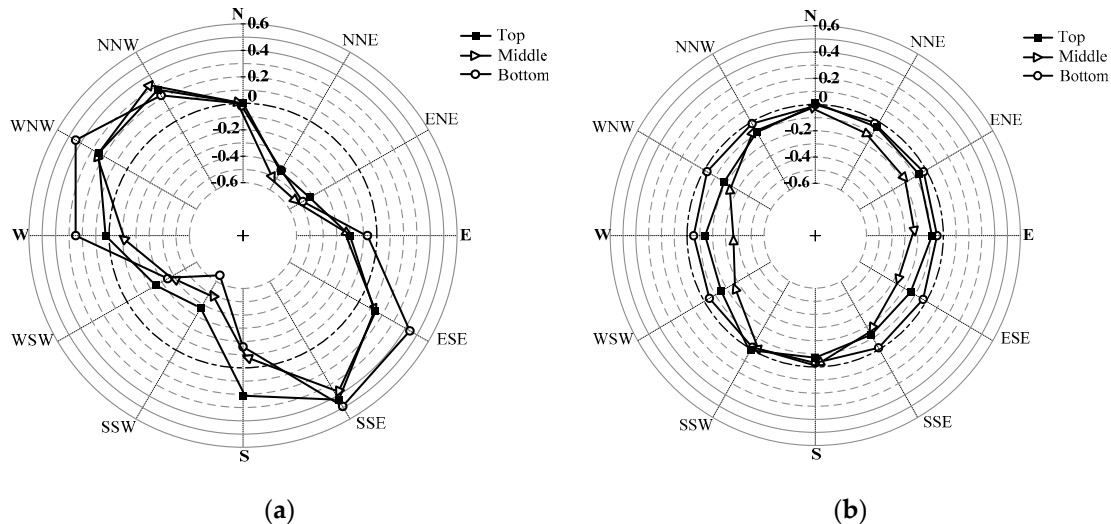

**Figure 12.** Distortions after (**a**) water cooling; (**b**) nitrogen cooling.

### 3.3. Mechanical Properties

The microstructures of THA obtained after nitrogen and water cooling are shown in Figure 13. It is observed from this figure that the tungsten grains are uniformly dispersed in a matrix alloy during both of the cooling processes. Moreover, no significant microstructural differences are observed.

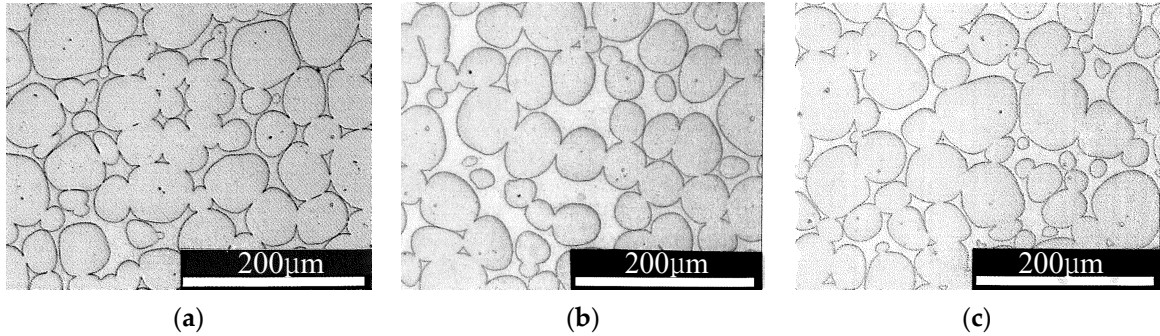

**Figure 13.** (**a**) Sintered microstructure before heat treatment; (**b**) heat-treated and water-cooled; (**c**) heat-treated and nitrogen-cooled.

The qualities of heat-treated parts in terms of mechanical properties such as, hardness, tensile strength, and elongation, were assessed. The results of this evaluation are summarized in Table 1. It is observed that the mechanical properties of the sintered tube have significantly improved after the heat treatment [50]. The differences in the mechanical properties with respect to water and nitrogen cooling processes are almost similar; however, it is observed that the nitrogen cooling process produces more or fewer ductile properties owing to the relatively lower cooling rate.

**Table 1.** Mechanical properties of sintered tube after heat treatment.

| Heat Treatment | Tensile Properties | | Hardness (HRC) |
|---|---|---|---|
| | $\sigma_t$ (kg/mm$^2$) | $e_t$ (%) | |
| Sintering | 80.4 | 5.3 | 26.5 |
| Sintering + water cooling | 96.6 | 22.9 | 29.7 |
| Sintering + nitrogen cooling | 95.1 | 24.8 | 27.9 |

### 3.4. Feasibility of Nitrogen Cooling

The feasibility of nitrogen cooling was evaluated in comparison with water cooling. In the case of conventional water cooling, the reduced yield ratio obtained due to the machining loss incurred by removing the oxidized surface is considered a drawback. Furthermore, the distortion of the workpiece during water cooling sometimes requires additional correction work. Non-uniform cooling caused by the gradual immersion of workpiece into the water pool can cause bending and twisting of the workpiece. This is particularly true for workpieces with a high length-to-diameter ratio.

The proposed nitrogen cooling system suggests a novel process for producing a clean part with less distortion. Table 2 compares costs for both nitrogen and water cooling. When compared to conventional water cooling in a pool, nitrogen cooling is an eco-friendly and cost-effective method due to the simplicity of the process. However, the cooling capacity of the nitrogen cooling process was relatively lower than that of water cooling. It was established that the cooling capacity of the nitrogen cooling process could be significantly increased by controlling the circulation speed and gas pressure.

**Table 2.** Comparison of costs for both nitrogen and water cooling.

| Item | Nitrogen Cooling (NC) | Water Cooling (WC) | Remarks |
|---|---|---|---|
| Equipment cost | Intermediate | Low | NC: needs vacuum chamber |
| Floor space | Low | Intermediate | WC: needs water reservoir |
| Operation cycle | Fast | Slow | - |
| Installation cost | Low | Very High | WC: water-proof design |
| Rearrangement cost | Low | Very High | WC: needs pit reconstruction |
| Eco friendly | High | Low | WC: contaminated sludge |
| Operating cost | Low | Low | |
| Maintenance cost | Very Low | Intermediate | WC: plumbing, leakage, water pumping |
| Post machining | Low | High | WC: low circularity |
| Yield ratio | High | Low | WC: large machining loss |

## 4. Conclusions and Future Work

An innovative nitrogen cooling furnace system was developed in this study for achieving a high-quality cooling process. In this process, the dry nitrogen in the cooling chamber exchanged heat with the workpiece it contacted, resulting in an accelerated cool-down. The heated nitrogen gas was forcibly circulated and transported to the chiller in a closed loop, where the heat was reduced, and the cooled gas was circulated again. The implementation tests using tubular THA confirmed that nitrogen gas cooling is an eco-friendly cooling medium, which can produce cleaner products. In addition, it was observed that this cooling system provided more uniform cooling and less distortion than water cooling. The cooling rate of nitrogen gas cooling is lower than that of water cooling; however, it was significantly enhanced by controlling the circulation speed and pressure. Future work concerns system upgrades to increase the cooling capacity of the nitrogen cooling system. Obviously, the optimal combination of gas pressure with circulation speed will be investigated since they have an important influence on the cooling capacity.

**Author Contributions:** Formal analysis: J.H.K.; investigation: Z.S.P.; methodology: J.K.; software: Y.Y.W.; supervision: Y.H.M.; validation: H.L.

**Funding:** This research was funded by National Research Foundation of Korea, grant number 12R1A5A1048294.

**Conflicts of Interest:** The authors declare no conflict of interest.

## Nomenclature

| | |
|---|---|
| $C_p$ | Specific heat, J kg$^{-1}$ °C$^{-1}$ |
| $D_i$ | Inner diameter of the tube, m |
| $D_o$ | Outer diameter of the tube, m |
| $e_t$ | Total elongation, % |
| $h$ | Heat transfer coefficient, W m$^{-2}$ °C$^{-1}$ |
| $k$ | Thermal conductivity, W m$^{-1}$ °C$^{-1}$ |
| $L$ | Length of the tube, m |

**(Greek symbols)**

| | |
|---|---|
| $\rho$ | Density, kg m$^{-3}$ |
| $\sigma_t$ | Tensile strength, kg mm$^{-2}$ |

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
