# Peer review of "Forced Circulation of Nitrogen Gas for Accelerated and Eco-Friendly Cooling of Metallic Parts"

_applsci, doi:10.3390/app9183679_

Round 1

Reviewer 1 Report

The paper is interesting by treating the well known advantages using the nitrogen in cooling tecniques. In my opinion, the authors should show the economical convenience when the cooling is obtained by means of the nitrogen instead of the water. Besides, the authors have to show how the heat transfer coefficients for the water and for the nitrogen have been calculated for the geometry investigated.

Author Response

Please refer to attached revision note.

Reviewer 2 Report

Please, have a look st the attached file for my comments 

Author Response

Please refer to the attached revision note. 

Round 2

Reviewer 1 Report

The authors have answered to my questions correctly. The paper is now acceptable as is, in my opinion.

Author Response

Thank you very much !

Reviewer 2 Report

Firstly I thank the Authors for considering my suggestions. Before the work can be published, some enhancements are compulsory:

1) all the units of measurements are incorrectly reported (e.g. J*kg^-1*°C*^-1 rather than J/(kg °C ) )

2) the units of measurements in Figure 10 are still incorrectly reported, despite my previous suggestion...

3) once again, the Introduction section is extremely poor. In this section, the Authors need to provide:

- sufficient background on the investigated topic

- an exhaustive assessment of the current status (this means that a sentence like “Several prior studies have researched the nitrogen cooling process [11-20].” cannot be considered acceptable)

- a clear statement regarding what knowledge gap (and how) their work will fill compared to the current status

Author Response

(The authors gave the same response as above.)

Round 3

Reviewer 2 Report

The paper can be published